# Annealing of Al-Zn-Mg-Cu Alloy at High Pressures: Evolution of Microstructure and the Corrosion Behavior

**DOI:** 10.3390/ma14082076

**Published:** 2021-04-20

**Authors:** Chuanjun Suo, Pan Ma, Yandong Jia, Xiao Liu, Xuerong Shi, Zhishui Yu, Konda Gokuldoss Prashanth

**Affiliations:** 1School of Materials Engineering, Shanghai University of Engineering Science, 333 Longteng Road, Shanghai 201620, China; s13115413752@163.com (C.S.); lx18845130767@163.com (X.L.); shixuer05@mails.ucas.ac.cn (X.S.); yu_zhishui@163.com (Z.Y.); 2Laboratory for Microstructures, Institute of Materials, Shanghai University, 99 Shangda Road, Shanghai 200444, China; 3Department of Mechanical and Industrial Engineering, Tallinn University of Technology, Ehitajate tee 5, 19086 Tallinn, Estonia; kgprashanth@gmail.com; 4Erich Schmid Institute of Materials Science, Austrian Academy of Sciences, Jahnstraße 12, A-8700 Leoben, Austria; 5CBCMT, School of Mechanical Engineering, Vellore Institute of Technology, Vellore 632014, India

**Keywords:** high-pressure annealing, Al-Zn-Mg-Cu alloy, microstructure, corrosion behavior

## Abstract

Extruded Al-Zn-Mg-Cu alloy samples with grains aligned parallel to the extrusion direction were subjected to high-pressure annealing. The effects of annealing pressure on the microstructure, hardness, and corrosion properties (evaluated using potentiodynamic polarization (PDP) and electrochemical impedance spectroscopy (EIS)) were investigated. Phase analysis showed the presence of MgZn_2_ and α-Al phases, the MgZn_2_ phase dissolved into the matrix, and its amount decreased with the increasing annealing pressure. The recrystallization was inhibited, and the grains were refined, leading to an increase in the Vickers hardness with increasing the annealing pressure. The corrosion resistance was improved after high-pressure treatment, and a stable passivation layer was observed. Meanwhile, the number of corrosion pits and the width of corrosion cracks decreased in the high-pressure annealed samples.

## 1. Introduction

Al-Zn-Mg-Cu-based alloys are widely applied in the aerospace and automobile sectors due to their excellent properties, such as high specific strength, formability, and corrosion resistance [1,2,3]. As a liquid-phase process, casting is capable of producing lightweight bulk components and is conventionally used to fabricate Al-Zn-based alloys [4,5]. Casting is a simple and relatively low-cost process. However, it may lead to large dendritic microstructures, shrinkage during solidification, etc. [6,7], which severely retard the application of this alloy. To improve the performance of Al-Zn-Mg-Cu-based alloys, strategies such as alloy modification [8,9] and homogenization treatment [10,11] have been adopted. According to our knowledge, the drawback of employing the conventional casting process is that the Zn content is restricted to 8 wt.% due to the presence of possible solidifying problems, such as microstructural coarsening, the macrosegregation of elements, and cracking [12,13,14].

As one of the advanced rapid solidification technologies, spray forming possesses the following advantages, i.e., increased solid solubility of alloy elements (due to rapid cooling), elimination of macrosegregation, reduced oxide content, and refined grains [15,16,17]. Currently, the spray forming technique is widely used to synthesize Al-Zn-Mg-Cu alloy, and the percentage of Zn element can be higher than 8 wt.%, unlike the conventional casting process [18]. For example, Li et al. prepared the Al-Zn-Mg-Cu-based alloy with a Zn content up to 11 wt.% by the spray deposition technique [19]. However, due to the characteristics of the spray deposition technique, a certain number of pores are inevitably introduced into the spray deposited alloy, which leads to crack initiation and subsequent failure of the component [20]. Extrusion is usually used to further increase the relative density of spray deposited alloys [14,21]. On the other hand, due to extrusion, the closed pores and the second phase are distributed in a chain along the extrusion direction to form a weak surface, and the alloy displays obvious structural anisotropy. Hence, further secondary processing is required to improve the properties of the Al-Zn-based alloys.

It is well known that the second phase particles [22], annealing temperature [23], and annealing time [24] are key parameters that can remarkably affect the recrystallization behavior. However, the effect of pressure on the recrystallization process is often undermined/overlooked. Recently, the use of high pressure on material processing has attracted considerable attention. The application of high pressure is characterized by a refined microstructure, increased chemical homogeneity, and the extension of solid solubility [25,26,27]. The material after extrusion may be annealed at high pressures to overcome the extrusion-based anisotropy. However, systematic work on such a topic has not been reported, and accordingly, in this study, the Al-Zn-Mg-Cu-based alloy prepared by spray deposition and extrusion was selected as raw material and was further subjected to high-pressure annealing. The effects of pressure on microstructure, mechanical properties, corrosion properties, and corrosion mechanisms were investigated systematically. The aim of this present study is to better understand the effects of annealing pressure on the recrystallization process along with the microstructural changes and corresponding properties of alloys.

## 2. Materials and Methods 

### 2.1. Materials and Experimental Procedure

In this study, a spray-deposited and -extruded (with an extrusion ratio 16:1) Al-Zn-Mg-Cu-based alloy of 20 mm diameter cylinders and 18 mm in length was supplied as the raw material. The nominal composition of the alloy used in the present study is listed in Table 1. The experiment was carried out using an HTDS-032F high-pressure six-sided top hydraulic press (Shenyang Scientific Instrument Co., Ltd., Chinese Academy of Sciences, Beijing, China). Six top hammers were moved up and down, left and right, and the cavity of the cube was squeezed to achieve quasi-static pressure. The pressures used were (a) atmospheric pressure, (b) 2 GPa, and (c) 3 GPa, respectively. The samples were heated in a graphite-based furnace inside the high-pressure solidification equipment. Pyrophyllite, the classic material, was used as an encapsulant and transmitting medium. In the experiment, the pressure was slowly increased up to the target value, and then the sample was heated to 833 K. The samples were kept at the target pressure and temperature for 15 min before being cooled down to room temperature by switching off the power supply. Finally, the samples were taken out for testing after releasing the pressure. A schematic diagram of high-pressure equipment and the cell assembly sample for high-pressure synthesis is shown in Figure 1.

### 2.2. Structural and Microstructural Analysis

The phase identification was carried out using a Rigaku D/max-2550 X-ray diffractometer (Rigaku Corporation, Tokyo, Japan) fitted with a monochromatic Cu-Kα radiation. The software JADE 6.5 was used for the analysis of the XRD data and calculation of the lattice parameters. Morphology and compositional homogeneity were examined using optical microscopy (OM) and scanning electron microscopy (SEM) after carrying out conventional polishing and etching. The etching was performed using Keller’s reagent (2 mL HF, 3 mL HCl, 5 mL HNO_3_, and 190 mL distilled H_2_O) for about 45 s. The volume fraction of the secondary phase was calculated using the Image-Pro Plus 6.0 software.

### 2.3. Corrosion Behavior and Microhardness

Both electrochemical impedance spectroscopy (EIS) and the potentiodynamic polarization (PDP) tests were conducted using an electrochemical workstation AUTOLAB PGSTAT302 (Metrohm, Herisau, Switzerland) and controlled by Nova 2.1 software. In this setup, the specimen was taken as the working electrode, the platinum sheet was used as the counter electrode, and the reference electrode used was a standard calomel electrode (SCE). The electrolyte used herein was quiescent 3.5 wt.% NaCl solution at ambient temperature. The samples were polished using standard metallographic practices. Above all, the open circuit potential (OCP) was measured for 30 min prior to the commencing of the polarization tests in order to achieve an approximately stable rest potential. Then, the impedance (Z) measurements were taken in the frequency range from 0.01 to 100 kHz with 10 mV set superimposed AC signal amplitude. Finally, each specimen was scanned at a rate of 1 mV·s^−1^ from the initial potential of −2 V, to a final potential of 1 V. The surface morphologies of the Al-Zn-Mg-Cu-based alloys after electrochemical experiments were investigated by using SEM to explore the corrosion behavior and to further understand the corrosion mechanisms. The microhardness test uses a Vickers hardness tester (Laizhou Huayin Test Instrument Co., Ltd., Yantai, China.) to test 10 points evenly on each sample. After removing the obtained maximum and minimum values, the average value was calculated as the microhardness of the sample.

## 3. Results and Discussion

### 3.1. Structural and Microstructural Characterization

Figure 2 shows the XRD patterns of the extruded Al-Zn-Mg-Cu-based alloys annealed at different pressures. As observed, the diffraction peak of MgZn_2_ phase was the strongest after extrusion, whereas after annealing at different pressures, all the peaks corresponding to the presence of MgZn_2_ phase decreased significantly in intensity. The peaks of (111)_α-Al_ changed from 38.440° of the alloy annealed under atmospheric pressure to 38.343° of annealed under 2 GPa; finally, the angle was 38.265° with increased annealed pressure, the lattice parameter of α-Al was calculated as 0.40176, 0.40669 and 0.41835 nm corresponding to annealing pressure from low to high, whereas the peak intensity of the (111)_α-Al_ increased remarkably.

The Arrhenius relationship (involving the pressure (*P*) and diffusion coefficient (*D*) of the atoms [28]) with Q as the activation energy for diffusion can be expressed as [29]:(1)$$D=D_{0}\cdot e^{-\left(Q/RT\right)}=D_{0}\cdot e^{-\left(PV/RT\right)}$$
where *R* is the gas constant, *T* is the temperature of the melt (K), *P* is pressure, and no change in *V*. When high pressure is applied (at the GPa level), the ratio between *D_p_* and *D*_0_ (i.e., the diffusion coefficients under high and atmospheric pressures) can be expressed as:(2)$$\frac{D_{p}}{D_{0}}=e^{\left(101,325-P\right)V_{0}/RT}$$

The *D_p_*/*D*_0_ was calculated to be 0.11 and 0.04 for the alloy annealed at 2 and 3 GPa, respectively. The diffusion coefficient decreased drastically with increasing annealing pressure. Therefore, the diffusion of the Mg and Zn atoms would be hindered, and the growth of the both the atoms (Mg and Zn) would be suppressed. Hence, more Mg and Zn atoms would dissolve into the α-Al matrix. When the pressure reached 3 GPa, almost all of the Mg and Zn dissolved into the α-Al matrix. The number and intensity of diffraction peaks of the MgZn_2_ phase disappeared at 3 GPa annealing pressure. Therefore, in this study, the solid solubility of Mg and Zn atoms increased with the increase in annealing pressure.

The optical microscopy images of the Al-Zn-Mg-Cu-based alloys are shown in Figure 3 and Figure 4. Figure 3 presents the microstructures of the extruded alloy. Figure 3a shows the microstructure along the extrusion direction; obvious directionality and elongated grains can be observed. Figure 3b shows the microstructure perpendicular to the extrusion direction; it can be seen that the grain boundaries were all irregular shape. Figure 4a shows the microstructure of alloy annealed under atmospheric pressure, where it is obvious that the grains were almost uniform equiaxed grains as a result of dynamic recrystallization. Figure 4b,c illustrate the microstructure of the alloy annealed under 2 and 3 GPa; it is clear that recrystallization grains formed, and most of the grain boundaries presented arched and serrated features. It can be seen that the grain boundaries of the alloy annealed at 2 GPa were sleeker than those of the alloy annealed at a pressure of 3 GPa.

At the same time, some larger grains were found in the samples annealed at a pressure of 3 GPa; there was a large size gap with some recrystallized grains. The partially recrystallized grains could be observed along with some larger grains that may not have undergone the recrystallization process. Based on these results, it can be inferred that incomplete recrystallization was observed in the samples annealed at 3 GPa pressure. In this case, the diffusion was severely suppressed, and the growth of the recrystallized grains was hindered, in the case that they were not fully grown. Secondary recrystallization occurred, and some large irregular grains showed secondary recrystallization. The grain size distribution of the alloy after annealing at different pressures is shown in Figure 5. The grain size of the alloy after atmosphere pressure annealing was generally larger, and the alloy after 2 GPa annealing decreased to some extent. The alloy after 3 GPa annealing was mainly composed of smaller grains; moreover, there were some larger secondary recrystallized grains. The average grain size for the Al-Zn-Mg-Cu samples was observed to be ~95, ~90, and ~72 μm, for the alloys annealed at atmospheric pressure, 2 GPa, and 3 GPa, respectively. The recrystallization phenomenon can be ascribed to the high stacking fault energies associated with Al and its alloys, which is prone to dynamic recovery during hot working [30,31,32,33].

The driving force for dynamic recrystallization is derived from the strain energy during plastic deformation, i.e., hot extrusion. The elongated deformed grains become equiaxed grains, eliminating lattice distortion in the structure, and simultaneously, the dislocations become annihilated (dislocation density reduces in number). Under high pressure, the solid solubility of solute atoms in the solvent increases, and hence, both the supercooling degree and the nucleation rate increase. According to the John Meir recrystallization grain size equation [34]:(3)$$d=K{\left(\frac{G}{\dot{N}}\right)}^{\frac{1}{4}}$$
where *d* is the average recrystallized grains diameter, $\dot{N}$ is the nucleation rate, *G* is the linear growth rate, and *K* is the proportionality constant. Obviously, the smaller the ratio, the finer the grains. The nucleation rate ($\dot{N}$) increased under high pressure, the interface energy decreased, and the growth rate (*G*) decreased, so the grain size (*d*) was reduced. Under the action of high pressure, the atomic diffusion/chemical potential gradient in the crystal decreased. The grain boundary was no longer the preferential channel for diffusion under high pressure. Bakker and Manning reached similar conclusions by studying the ordered binary alloys and the diffusion kinetics of atoms in crystals, respectively [35,36]. This also reduces the migration rate at the interface, and the high-pressure recrystallization grain growth rate is slower than that observed under atmospheric pressure. Hence, the higher the pressure during annealing, the stronger the barrier for grain boundary migration/diffusion; a smaller grain size leads to a lower rate of sub-grain boundary migration and grain growth. Krawczynska obtained similar results in the study of stainless steel [37].

The dislocation polygonalization occurred during the high-pressure annealing treatment, and the interface where the dislocation cells were located was destroyed to form sub-grain boundaries. At this time, the system produced a large number of sub-grains, resulting in an increase in the nucleation rate. In addition, the high-pressure suppressed the movement of dislocations, the interface migration rate, and the orientation difference between the small-angle sub-grain boundaries, resulting in slower grain growth rates. Lojkowski reached similar conclusions by studying the grain boundary migration mechanism of aluminum twin crystals and the influence of high pressure on the mobility of grain boundary atoms [38,39]. High pressures during annealing increase the free energy of the system; the intragranular diffusion increases; the grain boundary diffusion rate relatively decreases; and hence, the interface migration is again hindered [25,27,35,36,37,38,39,40]. Hence, under high-pressure annealing, a reduction in grain size was obtained, and substituting the parameters into the John Mayer equation, it may be evidently observed that the recrystallized grain size became smaller under annealing at high pressure.

Figure 6 shows the SEM images of the Al-Zn-Mg-Cu-based alloy annealed as a function of pressure. It can be clearly seen in Figure 6a that the white phase accumulated along the grain boundaries of the α-Al matrix. Combined with the XRD analysis, it can be confirmed that the white phase corresponded to the MgZn_2_ phase. The MgZn_2_ particles were unevenly distributed in the matrix, and the directionality caused by extrusion disappeared with annealing. Figure 6b,c show the microstructure of the alloy annealed at 2 and 3 GPa pressure, respectively. It can be observed from the microstructure that the directionality induced by the extrusion process still existed, and the MgZn_2_ phase was observed to be distributed along the extrusion direction. The area fraction of the MgZn_2_ phase was observed to be ~0.63% and ~0.40%, corresponding to Al-Zn-Mg-Cu alloy annealed under 2 and 3 GPa, respectively. The small broken MgZn_2_ phase was dissolved into the α-Al matrix, and the edges of the relatively large ones became passivated after high-pressure annealing. These results are consistent with the XRD data, where almost no peaks of the MgZn_2_ phase were observed.

The Vickers hardness of the Al-Zn-Mg-Cu-based alloys is shown in Table 2. Obviously, the hardness of the alloy after high-pressure annealing was much higher than its counterpart annealed at atmospheric pressure. In contrast to the alloy annealed at atmospheric pressure, the hardness of the alloy increased by ~34% and ~37%, respectively, with increasing annealing pressure to 2 and 3 GPa. The strengthening effect owing to the size reduction in MgZn_2_ along with solid solution strengthening contributed to such increment in the hardness values. Lang et al. reported a fine-grained Al-Zn-Mg-Cu alloy produced by strain-induced precipitation (a two-step deformation process), which exhibited significantly increased tensile ductility compared to the conventional hot-deformed alloy [31]. As analyzed in Section 3.1, the grain size decreased with increasing annealing pressure. The finer the grain size, the larger the grain boundary areas, leading to a more tortuous grain boundary path. A more torturous grain boundary path is not conducive to a further slip of dislocation, and hence improves the mechanical properties of these materials [32,33]. The movement of the dislocations was blocked along the grain boundaries, leading to dislocation pileups, where dislocation movement was hindered at the grain boundaries. With increasing applied stress, dislocation accumulation took place at the grain boundaries, and a larger number of dislocation pileups could be observed [5].

In addition, the solid solubility of both Mg and Zn in the α-Al matrix increased (Figure 4). Hence, it may be presumed that solid solution strengthening takes place when the Mg and/or Zn form a solid solution with the α-Al matrix, as the size and/modulus of the solute atoms may vary with the matrix, resulting in strain field variations. Local strain field variations are developed due to the presence of the precipitates, and they will readily interact with the dislocations. This can lead to the impedance of dislocation motion, leading to the increased strength of the material, thereby contributing to the increased hardness of the Al-Zn-Mg-Cu alloy [31,41].

### 3.2. Corrosion Behavior and Mechanisms

Figure 7 shows the Nyquist impedance plots of the Al-Zn-Mg-Cu-based samples annealed as a function of pressure in a 3.5% NaCl solution. The OCP of each sample was −0.802, −0.788, −0.643, and −1.0193 V. The equivalent circuit was fitted by NOVA2.1, and the analysis was executed with different numbers of circuit models. The minimum chi-square value was calculated for the best fit from the provided equivalent circuit model. The equivalent circuit was proposed using the impedance (Z) spectra. The Nyquist impedance plot along with the equivalent circuit model used for impedance data fitting of Al-Zn-Mg-Cu alloys is shown in Figure 8.

The Al-Zn-Mg-Cu-based alloys annealed under 2 and 3GPa produced similar Nyquist impedance plots. The medium–low-frequency capacitive reactance arc corresponds to the formation of the electric double layer between the oxide film and the solution, and the high-frequency capacitive reactance arc corresponds to the self-dissolving process of the oxide film. The Nyquist curve for Al-Zn-Mg-Cu alloys represents the two capacitive loops, and the diameter of the loop increases with increasing annealing pressure. A lower current density and higher resistance to corrosion are indicated by a bigger capacitive loop diameter [42].

An equivalent circuit, as shown in Figure 8, can express the interface state between the Al alloy substrate and the NaCl solution. The EIS results for the Al-Zn-Mg-Cu-based alloys are presented in Table 3, where *R*_1_ represents the charge transfer resistance, and a small *R*_1_ value indicates poor corrosion resistance (more corrosion). *R*_2_ is the inductor resistance, *Q* is the constant phase angle element, and *L* is the inductance. *C*_1_ and *C*_2_ correspond to the capacitance values of *Q*_1_ and *Q*_2_, respectively, and *N*_1_ and *N*_2_ are dispersion indices.

The charge transfer resistance of the alloy after extrusion was 7.5 kΩ, and samples annealed at atmospheric pressure, 2 GPa, and 3 GPa were 5, 25, and 35 kΩ, respectively. The increase in charge transfer resistance demonstrated the formation of a thick protective oxide layer on the sample surface that decreased the corrosion rate of the Al-Zn-Mg-Cu-based alloy. The charge transfer resistance increased with increasing annealing pressure, where the transmission was suppressed, and the charge transfer became very difficult. Accordingly, the oxide film integrity of the electrode surface became stronger and could not be easily damaged, thereby remarkably increasing its corrosion resistance. This is consistent with the previously obtained Nyquist impedance plots. Therefore, it can be deduced that the corrosion resistance of the Al-Zn-Mg-Cu-based alloy in 3.5 wt.% NaCl solution is improved with increasing annealing pressure.

The potentiodynamic polarization plots for the Al-Zn-Mg-Cu-based annealed under different pressures in 3.5 wt.% NaCl solution for about 50 min are shown in Figure 9. The four Tafel curves corresponding to the four different samples (extruded and annealed at atmospheric pressure, 2 GPa, and 3 GPa) show a similar trend. The results of polarization experiments of Al-Zn-Mg-Cu alloy in 3.5% NaCl solution are shown in Table 4, where E_corr_ is the corrosion potential, and I_corr_ is the corrosion current. In general, an excellent corrosion resistance was demonstrated by higher E_corr_ and lower I_corr_ values.

According to Figure 9 and A, from the perspective of corrosion thermodynamics, the lower the corrosion potential, the easier it is for the material to corrode. Hence, the alloy annealed at 3 GPa pressure showed more corrosion followed by the material annealed at 2 GPa pressure, and finally, the sample annealed at atmospheric pressure. From the perspective of corrosion kinetics, the larger the corrosion current, the higher the corrosion rate, where material annealed at 3 and 2 GPa pressure showed similar behavior, and the sample annealed at the atmospheric pressure showed the worst corrosion rate among the four. It may be observed from the polarization resistance (R_P_) that the sample annealed at 3 GPa pressure showed the highest resistance followed by the sample annealed at 2 GPa pressure, and the sample annealed at atmospheric pressure showed the least resistance. Aluminum alloy is a passivation alloy. The information of the passivation zone can be obtained from the polarization curve. When the voltage increased, the current density increased from almost stable to stable. The corresponding potential at this point is called the pitting potential or the breakdown potential (E_b_). The corresponding current is the breakdown current (I_pass_); higher Eb and lower I_pass_ indicate excellent corrosion resistance. In this study, the Eb of the alloy after high pressure annealing (−629 and −627 Ω/cm^2^) was lower than that of the extruded alloy (−781 Ω/cm^2^) and the atmospheric pressure annealed state (−776 Ω/cm^2^). At the same time, the I_pass_ of 3GPa (0.853 × 10^−5^ A/cm^2^) was smaller than that of 2 GPa (1.512 × 10^−5^ A/cm^2^). It can be considered that 3 GPa has the best corrosion resistance. In this study of electrochemical corrosion, the main factors that affected the corrosion performance of the alloy were the solid solubility of Mg, Zn atoms, and grain size. In the high-pressure annealed samples, the MgZn_2_ phase gradually dissolved in the Al matrix with increasing annealing pressure, suggesting that the reduction in the volume of the second phase led to an increase in the corrosion resistance of the alloy due to reduced potential difference between the matrix and the reinforcement [43].

The sample annealed under atmospheric pressure underwent complete recrystallization and hence exhibited a relatively larger grain growth after recrystallization as compared to the samples annealed at 2 and 3 GPa pressure. Relatively larger grains lead to accelerated corrosion. With increasing the annealing pressure, a large amount of the MgZn_2_ phase dissolved in the Al-matrix, and at the same time, complete recrystallization was not observed with the application of pressure. Hence, the subsequent grain growth was also avoided due to sluggish diffusion conditions, and a finer grain size was retained. Hence, for the alloy annealed with pressure, the corrosion resistance was observed to be better [44]. For high-pressure annealed alloys, the average grain size decreased with increasing pressure, and the corrosion resistance increased. The protective film formed on the metal surface was susceptible to permeation damage in the chloride ion (Cl^−^) environment, Cl^−^ adsorbed on the surface of passivation film, and the passivation process of the aluminum alloy was inhibited. The adsorption of Cl^−^ on the surface of the alloy caused the electric field effect and accelerated the dissolution rate of the metal surface. Moreover, the Cl^−^ formed a coordination compound with the metal successively, and the dissolution rate of the metal ion was accelerated. As aluminum alloy is an active metal, the surface of the sample was inclined to form a passive film after polishing and degreasing. However, the passivation film was not dense and was susceptible to Cl^−^ damage, causing Al to be exposed and to react with Cl^−^ [45,46,47].

According to these above analyses, the application of high pressure can improve the corrosion resistance of the Al-Zn-Mg-Cu alloy. On one hand, high-pressure treatment can improve relative density and reduce the number of pores in the Al-Zn-Mg-Cu-based alloy matrix, effectively prevent the penetration of Cl^−^ and other atoms into the surface of the sample, and reduce the corrosion rate [48]. On the other hand, the volume of the MgZn_2_ phase and the grain size after high pressure were reduced. The electrochemical corrosion caused by the local potential difference of the alloy due to the second phase was reduced, thereby reducing the overall corrosion process.

Figure 10 demonstrates the autocatalytic mechanism of pitting corrosion. Pitting corrosion is a corrosion phenomenon caused by the local accelerated dissolution of the protective passivation film of the alloy [49,50]. Generally, the initiation of pitting corrosion is related to the heterogeneity of the surface structure of the metal or the discontinuity of the passivation oxide film. For example, the initiation of pitting corrosion is normally related to inclusions, second phase particles, grain boundaries, defects, mechanical damage, or dislocation [51,52,53]. Existing studies have shown that in this aluminum alloy system, intermetallic compound MgZn_2_ will act as an anode particle with a higher potential than the substrate [54,55]. The intermetallic compound plays an important role in affecting the electrochemical corrosion process of alloys [56,57]. In this study, the second phase MgZn_2_ was first oxidized as the anode particle, and then resulted in a micropit, which initiated the pitting process [58]. Pitting corrosion is a self-catalytic process and will expand once initiated and change the local environment to further promote the growth of pitting corrosion. The relevant redox reactions during this process are described as follows: oxygen in water acts as a reducing agent; the reduction reaction that occurred outside the pitting pit is as follows:O_2_ + 2·H_2_O + 4·e^−^ → 4·OH^−^(4)

The oxidation reaction that occurred in the pit is as follows:Al → Al^3+^ + 3·e^−^(5)

With the increase in the number of metal cations (Al^3+^) in the pit, the resulting electric field promoted the movement of the number of anions (Cl^−^), forming aluminum chloride at the bottom, and maintaining electrical neutrality. Due to the hydrolysis of Al^3+^, the reaction equation can be given as:Al^3+^ + 3·H_2_O → Al(OH)_3_ + 3·H^+^(6)

It can be seen from Equation (6) that the increase in acidity at the bottom of the corrosion hole promoted a further development of corrosion.

Figure 11 depicts the surfaces of specimens after electrochemical corrosion tests. It can be seen that the Al-Zn-Mg-Cu alloy underwent local corrosion; the degree of corrosion was different in different samples annealed as a function of pressure. The most corrosion products were observed in the sample after extrusion; this means that it suffered the strongest corrosion, as shown in Figure 11a. A large number of studies have shown that the corrosion products are mainly Al(OH)_3_ [59,60,61,62]. The surface was uneven, and less corrosion products were observed in the sample that was annealed under atmospheric pressure, as shown in Figure 11b. The corrosion surfaces became smooth, and fewer corrosion products were observed on the surface of the samples annealed at 2 and 3 GPa atmospheric pressure Figure 11c,d. It is obvious that many coarse corrosion cracks were observed along the surface of the Al-Zn-Mg-Cu sample annealed under atmospheric pressure (Figure 12a). Moreover, severe corrosion at the grain boundaries could be observed. After high-pressure annealing treatment, the width of corrosion cracks decreased remarkably (Figure 12b–d), and at the same time, the corrosion at grain boundaries became weak.

Corrosion cracks promote the diffusion of ions through them, and the surficial corrosion layer cannot effectively protect the underlying metal. Hence, a further propagation of corrosion damage beyond the surface layer forming a partially corroded region was realized. The situation greatly improved after high-pressure annealing treatment, which indicates that the application of high pressure can effectively prevent the formation and growth of corrosion pits and microfractures.

As described in Section 3.1, the grains were dramatically refined after high-pressure treatment; meanwhile, more grain boundaries were generated. Wang et al. found that during the corrosion stage, grains predominantly bear the crack-driving force, and grain boundaries resist the microcrack propagation along or across the grain boundaries so that the corrosion rate is delayed [63]. The mechanism of electrochemical corrosion is that a corrosion potential exists (due to the potential difference between the intermetallic compounds and the Al-matrix), which leads to the formation of corrosion microcell, accelerated corrosion dissolution between intermetallic compounds and the Al-matrix. Combined with the previous analysis, the percentage of the second phase, such as η (MgZn_2_), had a considerable effect on the corrosion process, and the corrosion degree increased with the increase in the second phase content. The potential of the η phase (MgZn_2_) is negative compared to that of the Al matrix, so it will be dissolved as an anodic phase in the process of local corrosion of the alloy. Under the action of the active anion Cl^−^, the inherent equilibrium of the alloy is disturbed. In other words, the competitive adsorption of chloride ion and oxygen ion occurs, so that the chloride ion gradually replaces the original oxygen adsorption point on the metal surface [64,65]. It has been reported that more grain boundaries would facilitate the formation of an effective passive layer. This enhances the reaction of Al and O ions at the interface. Hence, the passage of metal ions from the surface is restricted towards the solution [66].

As a result, there is preferential pitting corrosion, and the etched nucleus can be produced at any part of the alloy surface (preferentially along with the surface defects). Intergranular deposition will be formed first. After pitting corrosion, intergranular corrosion will start, where regions with pitting corrosion act as the starting point. The intergranular corrosion follows the grain boundaries and gradually expands to all directions of the alloy. Based on the above analysis, high-pressure treatment can effectively reduce the corrosion rate and enhance the corrosion resistance of the alloy dramatically by (1) grain refinement and (2) solid solution strengthening.

## 4. Conclusions

The effect of annealing pressure on the microstructure and corrosion performances of the Al-Zn-Mg-Cu-based alloy was investigated; the following conclusions can be drawn:The Al-Zn-Mg-Cu alloy consists of α-Al and MgZn_2_ phases, and the grain size and the percentage of MgZn_2_ phase decrease with increasing annealing pressure. Moreover, the degree of recrystallization decreases with increasing annealing pressure.Strengthening due to grain refinement and solid solution strengthening leads to an increase in the Vickers hardness of the Al-Zn-Mg-Cu alloy from 132 (annealed at atmospheric pressure) to 180 HV (annealed at 3 GPa pressure).E_b_ increases and I_pass_ decreases with the increases in the annealing pressure. In addition, a higher impedance level of the Nyquist plot is the proof for the formation of a protective oxide layer on the surface of the Al-Zn-Mg-Cu-based alloy annealed under high pressures, suggesting that high-pressure annealing could effectively improve the corrosion resistance of the Al-Zn-Mg-Cu alloy.The scanning electron microscopy analysis reveals the presence of localized corrosion on the Al-Zn-Mg-Cu alloy, and it takes place mainly due to the formation of pits. The number of corrosion pits and the width of corrosion cracks decreases due to increased annealing pressure.

## Figures and Tables

**Figure 1 materials-14-02076-f001:**
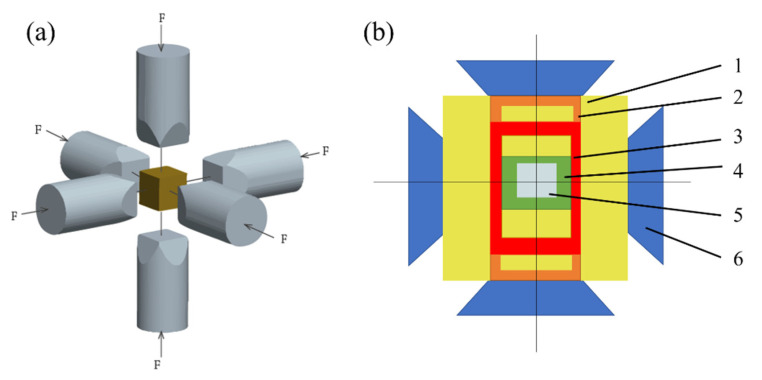
A schematic diagram showing (**a**) high-pressure equipment and (**b**) cell assembly sample for high-pressure synthesis: 1—pyrophillite, 2—conducting ring, 3—graphite crucible, 4—BN (clad), 5—sample, and 6—Top ram.

**Figure 2 materials-14-02076-f002:**
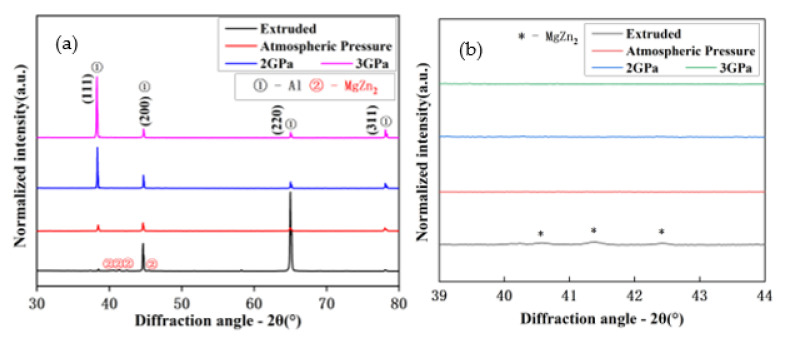
The X-ray diffraction patterns of the (**a**) extruded Al-Zn-Mg-Cu-based alloys annealed at different pressures (atmospheric pressure 1 and 2 GPa) and (**b**) plot showing the presence of MgZn_2_ phase.

**Figure 3 materials-14-02076-f003:**
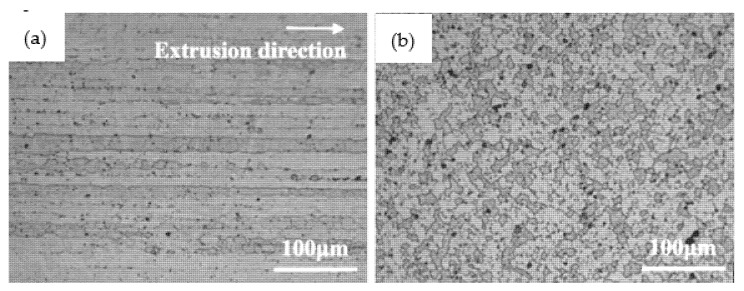
The optical microstructures of the extruded Al-Zn-Mg-Cu-based alloy: (**a**) along the extrusion direction and (**b**) perpendicular to the extrusion direction.

**Figure 4 materials-14-02076-f004:**
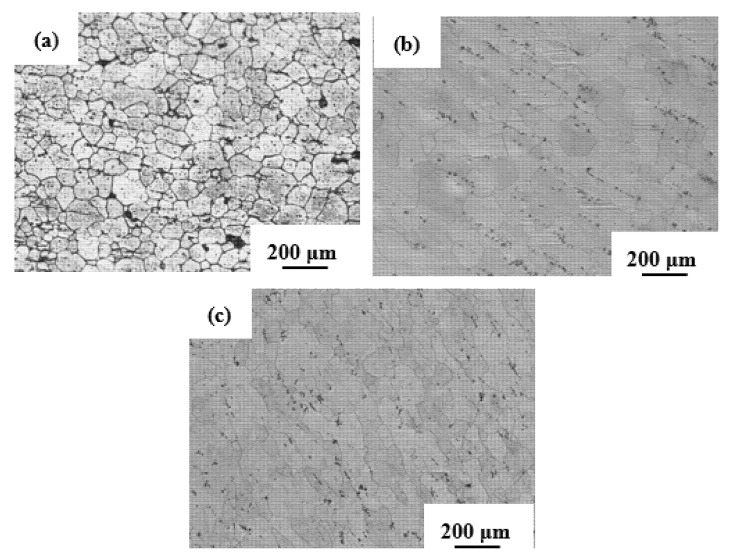
The optical microstructures of the extruded Al-Zn-Mg-Cu-based alloy annealed under (**a**) atmospheric pressure, (**b**) 2 GPa, and (**c**) 3 GPa, respectively.

**Figure 5 materials-14-02076-f005:**
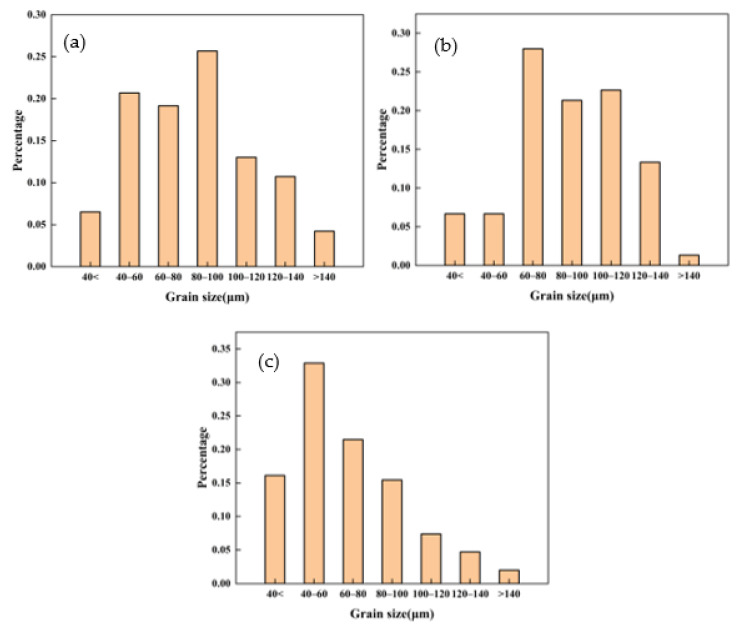
The grain size distribution statistics chart of the Al-Zn-Mg-Cu-based alloy annealed under (**a**) atmospheric pressure, (**b**) 2 GPa, and (**c**) 3 GPa, respectively.

**Figure 6 materials-14-02076-f006:**
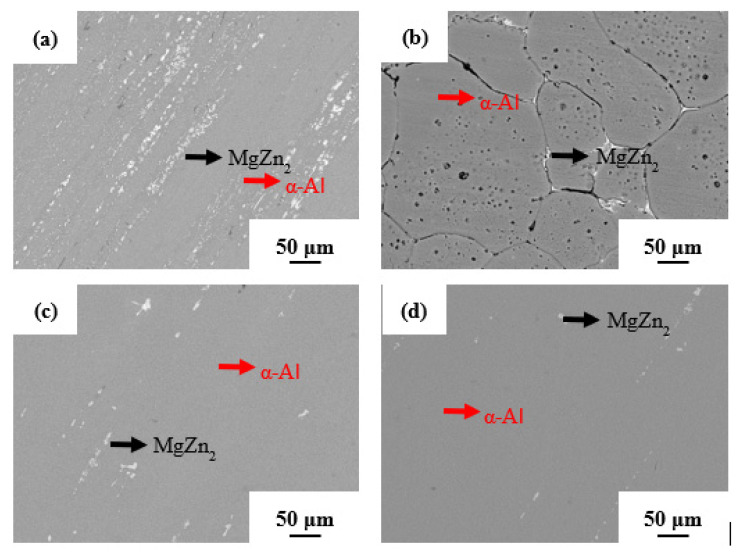
The scanning electron microscopy images of the (**a**) extruded Al-Zn-Mg-Cu-based alloy annealed under (**b**) atmospheric pressure, (**c**) 2 GPa, and (**d**) 3 GPa, respectively.

**Figure 7 materials-14-02076-f007:**
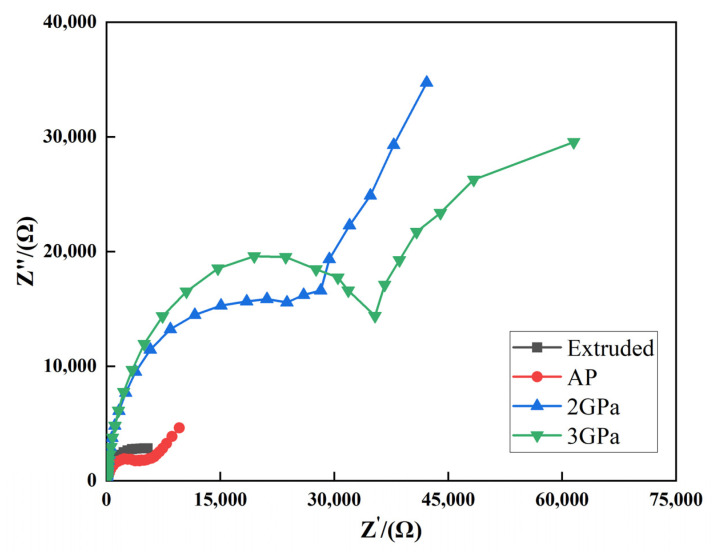
The Nyquist impedance plots for the extruded Al-Zn-Mg-Cu-based samples annealed as a function of pressure in 3.5 wt.% NaCl solution.

**Figure 8 materials-14-02076-f008:**
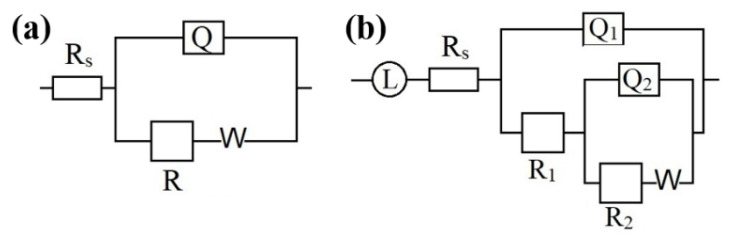
Equivalent circuit models used for impedance data fitting for the Al-Zn-Mg-Cu-based alloy after extrusion and annealing under (**a**) atmospheric and (**b**) 2 and 3 GPa.

**Figure 9 materials-14-02076-f009:**
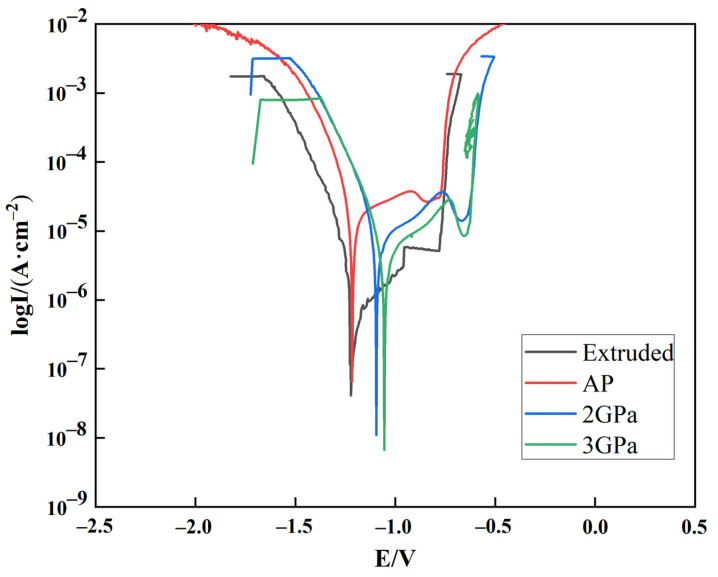
Potentiodynamic polarization plots for the extruded Al-Zn-Mg-Cu alloy annealed under different pressures.

**Figure 10 materials-14-02076-f010:**
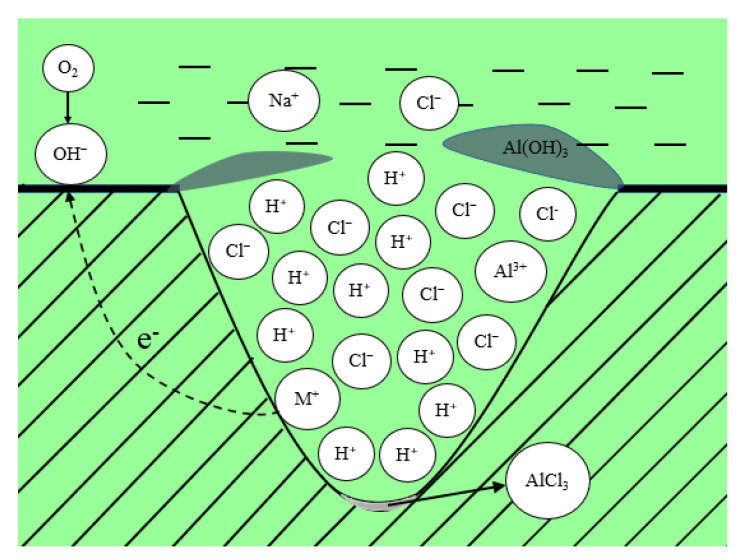
Autocatalytic mechanism of pitting corrosion.

**Figure 11 materials-14-02076-f011:**
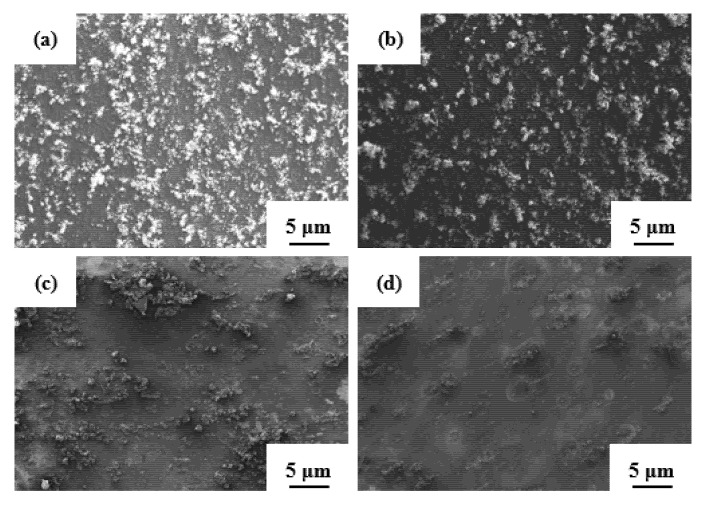
The scanning electron microscopy images of the (**a**) extruded Al-Zn-Mg-Cu samples annealed as a function of pressure and tested for corrosion in a 3.5 wt.% NaCl solution: (**b**) atmospheric pressure, (**c**) 2 GPa, and (**d**) 3 GPa, respectively.

**Figure 12 materials-14-02076-f012:**
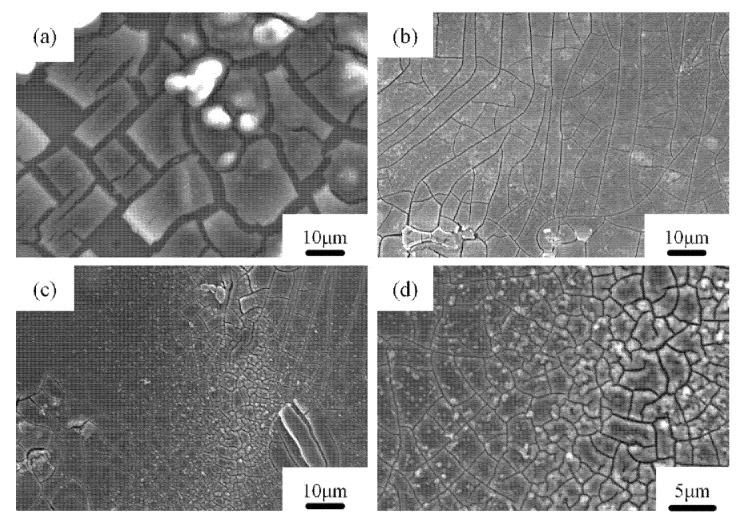
The scanning electron microscopy images of the Al-Zn-Mg-Cu alloy as a function of annealing pressure and after corrosion in 3.5 wt.% NaCl solution: (**a**) sample annealed at atmospheric pressure, (**b**) 2 GPa, and (**c**,**d**) 3 GPa, respectively.

**Table 1 materials-14-02076-t001:** Chemical composition of the Al-Zn-Mg-Cu alloy.

Elements	Zn	Mg	Cu	Al
Wt.%	10.81	2.10	1.52	Bal.

**Table 2 materials-14-02076-t002:** The Vickers hardness data observed for the Al-Zn-Mg-Cu-based samples.

Pressure	Extruded	Atmospheric Pressure	2 GPa	3 GPa
HV_0.2_	146 ± 3	131 ± 3	176 ± 2	180 ± 3

**Table 3 materials-14-02076-t003:** EIS fitting results for the extruded Al-Zn-Mg-Cu alloy annealed at different pressures.

Pressure	Extruded	Atmospheric Pressure	2 GPa	3 GPa
Circuit	a	a	b	b
R_s_/Ω	19.5	61.8	18.7	18.0
R_1_/Ω	7506.2	4923.2	25,298	34,458
R_2_/Ω	-	-	−1.1T	−1.1T
C_1_/μMho·s^N^	2.94	41	1.99	2.16
N_1_	0.7921	0.8052	0.9790	0.8564
C_2_/μMho·s^N^	-	-	23.7	37.9
N_21_	-	-	0.815	0.736
L_2_/μH	-	-	3.46	3.85

**Table 4 materials-14-02076-t004:** Results of the polarization experiments for the extruded Al-Zn-Mg-Cu alloy annealed at different pressures measured in 3.5% NaCl solution.

Pressure	I_corr_ (A/cm^2^)	E_corr_ (Ω/cm^2^)	R_p_ (Ω/cm^2^)	E_b_ (Ω/cm^2^)	I_pass_ (A/cm^2^)
Extruded	0.3024 × 10^−5^	−1209	2324	−781	3.039 × 10^−5^
Atmospheric pressure	0.1724 × 10^−5^	−1244	4640	−776	3.131 × 10^−5^
2 GPa	0.8127 × 10^−5^	−1094	5930	−629	1.512 × 10^−5^
3 GPa	0.5981 × 10^−5^	−1055	9225	−627	0.853 × 10^−5^

## Data Availability

Data may be available on request made to the corresponding author.

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
