# Peer review of "Annealing of Al-Zn-Mg-Cu Alloy at High Pressures: Evolution of Microstructure and the Corrosion Behavior"

_materials, 2021, doi:10.3390/ma14082076_

Round 1

Reviewer 1 Report

Review of paper no. materials-1166879-v1 titled Annealing of Al-Zn-Mg-Cu alloy at high pressures: Evolution of microstructure and corrosion behavior by C. Sun et al.

This is an interesting manuscript that studies the effect of high pressure annealing on microstructure and corrosion behavior of a quaternary aluminum alloy. It is found that the annealing causes a secondary phase dissolution in the matrix and improves the corrosion resistance. The paper is acceptable for publication subject to major revision.

1.A schematic of the high-pressure apparatus should be provided. It would help to specify the sample dimensions and mention the forces that were used to compact the sample.

2.To my knowledge, the atmospheric pressure is 101 325 Pa, i.e., 0.1 MPa and not 1 GPa, as mentioned by the authors on line 77. Please, correct your statement.

3.The XRD patterns given in Fig. 1 have a very low resolution. Please, enlarge the areas with secondary phase peaks. Miller indices should be also provided.

4.It is mentioned that “the lattice parameter of α-Al phase increases with high temperature annealing”. However, I do not see any Rietveld analysis of the data. Please, provide the lattice parameters.

5.Relationship (1) is valid for a molten alloy. Are you sure that the alloy was molten during annealing? The annealing was conducted at 560 °C (line 81). What is the melting point of the investigated alloy?

6.It is mentioned that the alloy annealed at atmospheric pressure “shows the worst corrosion rate among the three” (lines 288-289). However, the results given in Table 4 indicate the lowest corrosion rate. Please, explain your statement.

7.The polarization curves (Fig. 6) should be given in a full scale. It is very difficult to compare them if the image is cropped. Please, show the upper part.

8.The paper deals with pitting. It is, therefore, necessary to provide the pitting potentials and compare them.

9.The paper discusses the pitting corrosion of Al alloys (Fig. 7). The effect of intermetallic compounds on pitting should be also considered and recent papers should be referenced (Journal of The Electrochemical Society, 165 (2018) C807-C820; Materials 12 (2019) 1661; etc.).

10.Have you measured an open circuit potential (OCP) of the alloy?

11.Which corrosion products were formed on the alloy substrates? It would help to include a chemical analysis (EDS) in the paper.

12.It is mentioned that “the potential of the η-phase (MgZn2) is negative compared to the Al matrix”. A reference is required to confirm the claim.

13.In conclusion 3 you state that Eb and Ipass decrease with increasing annealing pressure. However, the parameters have not been provided in the paper. Please, include them in Table 4.

14.Please, pay attention to indices. Al3+ should be given as Al3+ (line 347), MgZn2 as MgZn2 (lines 294, 387, 389), etc.

End of comments

Author Response

See the file attached

Reviewer 2 Report

The article is devoted to the study of the influence of annealing modes at high pressure on the structure, mechanical properties, and corrosion behavior of an alloy of the Al-Zn-Mg-Cu system. The work presents interesting and new scientific results, however, there are quite a lot of inaccuracies and speculations in the work. Authors should pay attention to the following issues:

  1. The authors should add the structure and properties of the original extruded alloy to compare the changes in the structure after annealing.
  2. The diffraction patterns in Figure 1 should be enlarged and an extruded state should be added. In the diffraction pattern of the state after annealing at atmospheric pressure, a slight texture is observed. The authors should indicate the direction of measurement. In the context of texture, the following sentence needs to be commented: Line 118 “Whereas, the peak intensity of the α-Al phase increases.”
  3. In equation 1, it is necessary to comment on what the authors mean by “η0 is the viscosity of the melt at atmospheric pressure” “V0 is the initial volume of a liquid”. The authors did not heat the alloy above the solidus temperature.
  4. It is not clear how the pressure difference is related to the solubility of Mg and Zn in the aluminum solid solution. Lines 133-139
  5. The microstructure of extruded state in different directions should be added to Fig.2
  6. Lines 150-160. The authors point out the presence of large grains in the structure after annealing at 3 GPa and attribute this to incomplete recrystallization. This conclusion is not clear, and it is not clear why the authors do not consider the possibility of undergoing grain growth after recrystallization.
  7. Lines 171-179. This discussion looks speculative and should be supported by experiments or references.
  8. Lines 200-205. The obtained values of the fraction of the second phase should be analyzed in comparison with the equilibrium state phase diagram.
  9. Line 210 “Strengthening due to grain refinement….” Based on the results obtained, the grain size weakly depends on the processing mode; most likely, the main share is made by solid solution hardening, and the size of the MgZn2
  10. Conclusion 1 is not supported by the text of the article

The article requires major revision before publication in the journal

Author Response

See the file attached

Reviewer 3 Report

Notes:
1. For a better understanding of the method of annealing at controlled pressure, I recommend that you give a photo, drawing or diagram of the installation based on a hydraulic press, which was used for research.
2. In the description of the experimental research methodology, it is not indicated whether there was a preliminary deformation of the annealed alloy, and if so, what is its value. It is also necessary to bring the initial (before annealing) microstructure of the alloy, since in the study of recrystallization processes the initial structure (state) is of significant importance.
3. Row 157 gives the results for the average grain size at the pressures under study. The value of the average size does not give an understanding of the magnitude and degree of graininess of the microstructure. A histogram of the grain size distribution in the test sample should be presented. In turn, visually analyzing the microstructure shown in Figure 2, there is a feeling that at a pressure of 2 GPa, the coarse-grained microstructure of the three given is formed. Table 4 shows the results of determining the corrosion rate. From these results it follows that the 2 GPa sample has the highest corrosion rate, which is consistent with my assumption.
4. In Figure 3 it is necessary to indicate the phases.
5. Table 2 shows the results of measuring the Vickers hardness, but the method does not say about it. It is necessary to add a description of the hardness tests to the procedure and indicate how many measurements and in which part of the sample they were made.
6. The above remarks cast doubt on the first and second conclusions.

Author Response

See the file attached

Round 2

Reviewer 1 Report

Authors sufficiently adressed my comments. The paper is acceptable for publication.

Author Response

Thanks for the reviewer's comments.

Reviewer 2 Report

The authors have significantly revised the article, however, before publication, the following points need to be corrected:

  1. Lines 125-129: The authors should add to the methodology section, the method for determining the lattice period with such accuracy and the measurement error.
  2. 5. The quality of the drawing needs to be improved. It is practically unreadable.

Author Response

Point 1: Lines 125-129: The authors should add to the methodology section, the method for determining the lattice period with such accuracy and the measurement error.

Response 1: Revised and added in line 92-93.

Point 2: 5. The quality of the drawing needs to be improved. It is practically unreadable.

Response 2: Revised.

Reviewer 3 Report

The above remarks have been corrected, the article has been revised.

Author Response

Thanks for the reviewer's comments.